# Expanding the Colorectal Cancer Biomarkers Based on the Human Gut Phageome

Siyuan Shen,[a] Dongxue Huo,[a] Chenchen Ma,[a] Shuaiming Jiang,[a] ⊙ Jiachao Zhang[a,b]

[a]College of Food Science and Engineering, Hainan University, Haikou, China
[b]Key Laboratory of Food Nutrition and Functional Food of Hainan Province, Haikou, China

**ABSTRACT** With the increasing prevalence of colorectal cancer (CRC), extending the present biomarkers for the diagnosis of colorectal cancer is crucial. Previous studies have highlighted the importance of bacteriophages in gastrointestinal diseases, suggesting the potential value of gut phageome in early CRC diagnostic. Here, based on 317 metagenomic samples of three discovery cohorts collected from China (Hong Kong), Austria, and Japan, five intestinal bacteriophages, including *Fusobacterium nucleatum*, *Peptacetobacter hiranonis,* and *Parvimonas micra* phages were identified as potential CRC biomarkers. The five CRC enriched bacteriophagic markers classified patients from controls with an area under the receiver-operating characteristics curve (AUC) of 0.8616 across different populations. Subsequently, we used a total of 80 samples from China (Hainan) and Italy for validation. The AUC of the validation cohort is 0.8197. Moreover, to further explore the specificity of the five intestinal bacteriophage biomarkers in a broader background, we performed a confirmatory meta-analysis using two inflammatory bowel disease cohorts, ulcerative colitis (UC) and Crohn's disease (CD). Excitingly, we observed that the five CRC-enriched phage markers also exhibited high discrimination in UC (AUC = 78.02%). Unfortunately, the five CRC-rich phage markers did not show high resolution in CD (AUC = 48.00%). The present research expands the potential of microbial biomarkers in CRC diagnosis by building a more accurate classification model based on the human gut phageome, providing a new perspective for CRC gut phagotherapy.

**IMPORTANCE** Worldwide, by 2020, colorectal cancer has become the third most common cancer after lung and breast cancer. Phages are strictly host-specific, and this specificity makes them more accurate as biomarkers, but phage biomarkers for colorectal cancer have not been thoroughly explored. Therefore, it is crucial to extend the existing phage biomarkers for the diagnosis of colorectal cancer. Here, we innovatively constructed a relatively accurate prediction model, including: three discovery cohorts, two additional validation cohorts and two cross-disease cohorts. A total of five possible biomarkers of intestinal bacteriophages were obtained. They are *Peptacetobacter hiranonis* Phage, *Fusobacterium nucleatum animalis 7_1* Phage, *Fusobacterium nucleatum polymorphum* Phage, *Fusobacterium nucleatum animalis 4_8* Phage, and *Parvimonas micra* Phage. This study aims at identifying fine-scale species-strain level phage biomarkers for colorectal cancer diseases, so as to expand the existing CRC biomarkers and provide a new perspective for intestinal phagocytosis therapy of colorectal cancer.

**KEYWORDS** metagenome, colorectal cancer, bacteriophage, biomarkers

Worldwide, by 2020, colorectal cancer (CRC) has become the third most common cancer after lung and breast cancer (1, 2). Numerous researches have proved that colorectal cancer is closely related to human intestinal microorganisms, but the current research on cancer microbiome is almost only concerned with bacteria (3, 4), and rarely on bacteriophages (5). The individuals of bacteriophage are more than bacteria in microbiota, and bacteriophages are strictly host-specific, which are used to recognize the

Address correspondence to Jiachao Zhang, zhjch321123@163.com.

The authors declare no conflict of interest.

Microbiology Spectrum

disease characteristics in some diseases (6). Accordingly, phages have been shown to play a key role in the pathogenesis of colorectal cancer, suggesting the potential value of virus group detection in early disease screening (7). However, research on intestinal bacteriophages in CRC patients was limited, only one study explored the gut phage biomarkers at the genera level (8, 9).

To address these challenges, we used the Gut Phage Database (GPD), the most comprehensive and human intestinal phage gene database so far, for gut phageome annotation. Then, we performed a meta-analysis on the data sets of three cohorts, and validated biomarkers in two additional cohorts and two cross disease cohorts, involving 561 fecal metagenomes. Here, we aim to identify fine-scale species-strain level bacteriophage biomarkers for colorectal cancer disease, so as to expand the existed CRC biomarkers and rebuild the more accurate predictive models, which also provide a new view for CRC gut phagotherapy.

## RESULTS

**Alteration of the intestinal phageome in CRC patients.** Here, we collected 317 samples (Control, $n = 157$; CRC, $n = 160$) from Austria, China (Hong Kong) and Japan as the discovery cohort. We performed a comparative analysis of the intestinal phage alpha diversity and beta diversity between CRC patients and healthy subjects in the discovery cohort. The microbial phage alpha diversity of CRC group was significantly ($P < 0.05$, Wilcoxon rank-sum tests) lower than that of control group (Fig. 1A), which indicated that the intestinal phage community richness of CRC patients was higher than that of healthy subjects. At the same time, beta diversity analysis showed that there were separate clusters of control and CRC (Adonis, $P < 0.0001$, Wilcoxon rank-sum tests, Fig. 1B), which suggested that the intestinal bacteriophages of colorectal cancer patients were different from those of healthy subjects.

**Fine-scale bacteriophagic biomarkers identification for CRC disease.** When we observed that there were differences in the diversity and structure of intestinal bacteriophages in the control and CRC group, we were eager to know what might be the cause of these differences. Therefore, we further discovered 916 intestinal bacteriophages in the Hong Kong cohort ($P < 0.0001$), 135 intestinal bacteriophages in the Japanese cohort ($P < 0.0001$) and 2389 intestinal bacteriophages in the Austrian cohort ($P < 0.0001$) by using Wilcoxon Rank-Sum tests. Based on the differences in intestinal bacteriophages discovered in each discovery cohort, we were surprised to find that there were 18 bacteriophages in common between the China (Hong Kong) cohort and the Japan cohort ($P < 0.0001$), and 14 bacteriophages in common between China (Hong Kong) cohort and the Austria cohort ($P < 0.0001$). Even more exciting, we further identified five biomarkers of intestinal bacteriophages that were common to all three discovery cohorts with significant differences in control and CRC ($P < 0.0001$), which were, respectively, *Peptacetobacter hiranonis* Phage (ERR1018254_84 length _81930_VirSorter_cat_2), *Fusobacterium nucleatum animalis 7_1* Phage (ERR209701_284 length _46639_VirSorter_cat_2), *Fusobacterium nucleatum polymorphum* Phage (ERR1018241_502 length _44581_VirSorter_cat_2), *Fusobacterium nucleatum animalis 4_8* Phage (SRR1159789_2 length _65902_VirSorter_cat_2), and *Parvimonas micra* Phage (VIRSorter_NZ_DS483517_1_Parvimonas_micra_ATCC_33270). In the discovery cohort, the five markers achieved an area under the receiver-operating characteristic curve (AUC) of 0.8616 for the classification. The accuracy of phage biomarkers was similar to that of bacterial biomarkers in the same cohort (AUC = 88.58%), but the number of identified biomarkers was only 5, far less than the number of bacterial biomarkers. At the same time, it was exciting that not only the five bacteriophages biomarkers were significantly enriched in the CRC ($P < 0.0001$), but more importantly, the obtained *Fusobacterium nucleatum* Phage and *Parvimonas micra* Phage were consistent with the standard biomarkers of bacteria in CRC, *Fusobacterium nucleatum* and *Parvimonas micra*. In addition, the three intestinal bacteriophages biomarkers were all *Fusobacterium nucleatum* subspecies phages (Fig. 1C, E, D, F). Subsequently, in order to verify the accuracy of the five intestinal bacteriophage biomarkers we explored, we selected two countries with highly different geographical environments and dietary habits, one Asian country (Hainan, China) and one European

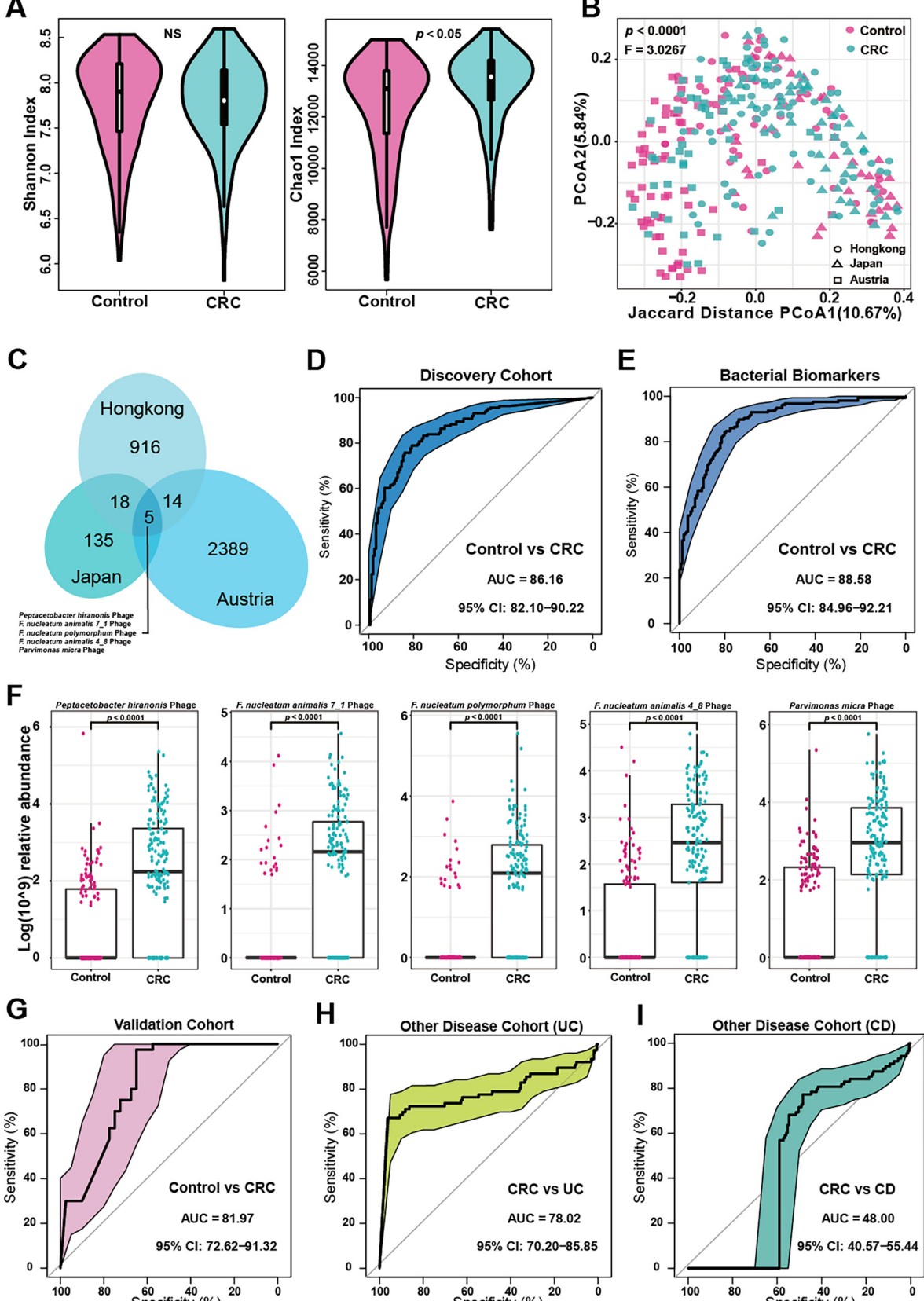

**FIG 1** Construction and validation of a prediction model for colorectal cancer(CRC) in bacteriophages. (A) Shannon and Chao1 Index shows alpha diversity between CRC ($n$ = 160) and control ($n$ = 157)(NS, $P$ = 0.27). (B) Principal coordinates analysis of Jaccard distance shows the

country (Italy), as our validation cohorts. Therefore, we used a total of 80 samples (Control, $n = 40$; CRC, $n = 40$) from China (Hainan) and Italy as validation cohorts. The method to verify the accuracy of the five intestinal phage biomarkers was as follows: receiver operating characteristic curve (ROC). In the validation cohort, the markers achieved an area under the AUC of 0.8197 for the classification, which indicated that the five phage biomarkers have preferable accuracy (Fig. 1G).

**Specificity of bacteriophagic diagnostic markers of CRC against other disease cohorts.** After we found five possible biomarkers for control and CRC. We wanted to test whether these five biomarkers could not only distinguish between control and CRC, but also differentiate between different diseases. Inflammatory bowel disease (IBD) includes ulcerative colitis (UC) and Crohn's disease (CD). In recent years, through in-depth studies by digestive experts from all over the world, the medical community has found that there is a certain relationship between inflammatory bowel disease and colorectal cancer: the risk of CRC in IBD is two to four times that of the normal population, and about 20% of patients with IBD develop CRC within 10 years of the onset (10). Meanwhile, there are also studies to prove that IBD and CRC have a strong correlation with intestinal microorganisms (11). Therefore, the selection of UC and CD disease groups can better reflect the differentiation effect of the five intestinal bacteriophage biomarkers. So, we did validation trials of bacteriophages biomarkers in the UC cohort and CD cohort. The UC cohort included 78 samples and the CD cohort included 88 samples. Excitingly, we observed that the five CRC-enriched phage markers also exhibited high discrimination in UC (AUC = 78.02%), which indicated that the five intestinal bacteriophages markers have excellent specificity in CRC disease (Fig. 1H). Unfortunately, the resolution of the five CRC phage biomarkers in CRC and CD was low (AUC = 48.00%), which may indicate that there is no specificity for CD and CRC, the five phage biomarkers (Fig. 1I).

## DISCUSSION

A growing number of studies have shown that bacteriophages can affect human health and disease states (12). However, until now, the role of intestinal phages in disease has been largely unexplored, which may be due to the lack of well-characterized reference genomes and phage databases with large amounts of data (13). For this reason, in this study, we used the Gut Phage Database, the most comprehensive and complete human intestinal phage gene database so far, and a total of 142,809 phages were annotated.

In previous studies, 22 virus genera were analyzed through model construction and could be used as markers to distinguish CRC from the control group (8), but no possible species-level biomarkers have been analyzed. It is well known that due to the huge individual differences between populations, the study of the association between diseases and microbiome is extremely complex, so the representativeness and accuracy of biomarkers at the generic level are far from enough. Therefore, more identifiable biomarkers are strongly needed to build more accurate prediction and diagnostic models. Only with more accurate biomarkers and more effective predictive models can follow-up treatment be better targeted.

Here, we innovatively constructed a relatively accurate prediction model, including: three discovery cohorts, two additional validation cohorts and two cross-disease cohorts. A total

**FIG 1** Legend (Continued)

stratification of CRC ($n = 160$) from control ($n = 157$) samples by bacteriophagic compositional profile of the discovery cohort and the *P value* represents the significance between the two groups (Wilcoxon rank-sum tests). (C) For discovery cohort, there are five bacteriophagic biomarkers with significant differences ($P < 0.0001$, Wilcoxon rank-sum tests) in China (Hong Kong, CRC, $n = 74$; Control, $n = 54$), Japan (CRC, $n = 40$; Control, $n = 40$) and Austria (CRC, $n = 63$; Control, $n = 46$). (D) In the discovery cohort, the bacteriophage biomarkers achieved an area under the receiver-operating characteristic curve (AUC) of 0.8616 for the classification. (E) In the discovery cohort, bacterial markers had an area of 0.8616 under the receiver-operating characteristic curve (AUC) for the classification. (F) The Log (10^9) relative abundance of the five bacteriophagic biomarkers in CRC ($n = 160$) and control ($n = 157$) are significantly different ($P < 0.0001$, Wilcoxon rank-sum tests), and the relative abundance of the bacteriophagic biomarkers in the CRC group is higher. (G) In the validation cohort (CRC, $n = 40$; control, $n = 40$), the markers achieved an area under the receiver-operating characteristic curve (AUC) of 0.8197 for the classification. (H) In the other disease cohort, the five biomarkers of CRC ($n = 160$) and ulcerative colitis (UC, $n = 76$) were classified with an area under the receiver-operating characteristic curve (AUC) of 0.7802. (I) In the other disease cohort, the five biomarkers of CRC ($n = 160$) and Crohn's disease (CD, $n = 88$) were classified with an area under the receiver-operating characteristic curve (AUC) of 0.4800.

**TABLE 1** Fecal metagenomic data included in this meta-analysis

| Cohorts | No. of cases | No. of controls | Accession |
|---|---|---|---|
| Discovery cohorts | | | |
| China (Hong Kong) | 74 | 54 | PRJEB10878 |
| Japan | 40 | 40 | DRA006684 |
| Austria | 46 | 63 | ERP008729 |
| Validation cohorts | | | |
| China (Hainan) | 8 | 12 | PRJNA663646 |
| Italy | 32 | 28 | SRP136711 |
| Other disease cohorts | | | |
| IBD-UC | 76 | | PRJNA400072 |
| IBD-CD | 88 | | PRJNA400072 |

of five possible biomarkers of intestinal bacteriophages were obtained. It is exciting that among the five intestinal bacteriophage biomarkers we obtained, three intestinal bacteriophages were labeled at the subspecies level of *Fusobacterium nucleatum* (*Fusobacterium nucleatum animalis 7_1*, *Fusobacterium nucleatum polymorphum,* and *Fusobacterium nucleatum animalis 4_8*), and one was labeled as *Parvimonas micra* phage. It is well known that *Fusobacterium nucleatum* and *Parvimonas micra* are standard diagnostic biomarkers of bacteria in colorectal cancer (14). Because phages are viruses that attack bacteria and are strictly host-specific. Therefore, based on the specificity of this host may give us more accurate biomarkers. According to the above conclusions, we may provide more accurate and targeted species-level biomarkers for the follow-up treatment of colorectal cancer. Here, we aim to identify fine-scale species-strain level bacteriophage biomarkers for colorectal cancer disease, so as to expand the existed CRC biomarkers and rebuild the more accurate predictive models, which also provide a new view for CRC gut phagotherapy.

## MATERIALS AND METHODS

**Sequence data collection.** Fecal metagenomic data for CRC and control were collected for the meta-analysis. For discovery cohorts, raw SRA files and sample information were downloaded from NCBI. In the NCBI, accession of China (Hong Kong) (15) is PRJEB10878, CRC, $n = 74$; Control, $n = 54$. In the NCBI, accession of Japan (16) is DRA006684, CRC, $n = 40$; Control, $n = 40$. In the NCBI, accession of Austria (17) is ERP008729, CRC, $n = 46$; Control, $n = 63$ (Table 1). For validation cohorts, raw SRA files and sample information were downloaded from NCBI. In the NCBI, accession of China (Hainan) (18) is PRJNA663646, CRC, $n = 8$; Control, $n = 12$. In the NCBI, accession of Italy (19) is SRP136711, CRC, $n = 32$; Control, $n = 28$ (Table 1). At the same time, the SRA files and sample information for the other disease validation cohorts we used were also downloaded from NCBI. Other disease cohorts included UC (20) ($n = 76$), which has been Accession PRJNA400072, and CD (20) ($n = 88$), which has been Accession PRJNA400072 (Table 1).

**Data quality control and phage database acquisition.** Whole-genome shotgun sequencing of the samples from all cohorts was carried out on Illumina HiSeq 2000/2500 platform with similar sequencing depths. The abundances of all samples were determined by aligning the reads to the Gut Phage Database (21). The Gut Phage Database we used is a database of 142,809 human intestinal phage genomes obtained by analyzing 28,060 human intestinal metagenomes and 2,898 reference genomes of intestinal bacteria around the world. The database is linked to: http://ftp.ebi.ac.uk/pub/databases/metagenomics/genome_sets/gut_phage_database/ using Bowtie2 (22). Subsequently, for any sample N, we calculated the relative abundance as follows:

Step: Calculation of relative abundance of phages in sample N

$$a_i = \frac{b_i}{\sum_i b_i}$$

$a_i$: the relative abundance of phages in sample N.
$b_i$: the number of mapped reads of phage i from sample N.

**Batch effect correction.** After obtaining the abundance tables of all the samples in the different cohorts, we used the online tool "BatchServer" for the samples in different cohorts to remove the batch effect. This online tool (https://lifeinfo.shinyapps.io/batchserver/) is based on the inside of the SVA in R software package ComBat function to remove batch effect (23).

**Selection of bacteriophage biomarkers and application of machine learning.** Five CRC bacteriophage biomarkers were identified using the random forest (RF) model (24) and the Wilcoxon Rank-sum test. We used the random forest model to search for biomarkers from 142809 intestinal phages and applied the R package "Ranger" (V0.12.1) to implement the random forest algorithm for each classification task. All the hyperparameters were set as default except for the number of trees set to 5000. The predictive performance of the RF model was evaluated by the cross-validation method 10-fold, and five bacteriophage biomarkers

with the contribution rate >0.1% were identified. At the same time, we used the Wilcoxon Rank-sum test to search for phages with significant difference ($P < 0.0001$) between CRC patients and healthy people in three discovery cohorts, and combined analysis was performed on the differential phages found in three discovery cohorts. Five phages were found that were significantly different in all three cohorts and were enriched in the intestinal tract of CRC patients. Interestingly, the five biomarkers found by the random forest model were the same as the five biomarkers found by the Wilcoxon Rank-sum test. Therefore, we set these five phages as biomarkers, and their AUC reached 86.16%.

**Acquisition of bacterial biomarkers and accuracy.** The bacterial species and abundance of the discovery queues were calculated using MetaPhlAn 2.0 software (25). Then we applied R package "Ranger" (V0.12.1) to realize the random forest algorithm for each classification task and used the random forest model to obtain biomarkers of bacterial CRC. All hyperparameters are set to default values except for the number of trees set to 5000. The prediction performance of RF model was evaluated by the cross-validation method 10-fold method, and 182 bacterial biomarkers with high contribution rate were screened out, and their AUC reached 88.58%.

**Statistics statement.** All statistical analyses were performed using R software. Vioplot was shown by the "vioplot" package. PCOA analysis was performed using the "ade4" package in R. The differential abundances of various profiles were tested with the Wilcoxon rank-sum test and were considered significantly different at $P < 0.05$. Boxplot was shown by the "ggplot2" package. ROC analysis was used to assess the performance of the microbial biomarkers using the "pROC" package in R. The Venn diagram was built using an online tool called "Omicstudio."

**Data availability.** The raw SRA files and sample information used in this article were downloaded from NCBI using the following accessions: PRJEB10878for China (Hong Kong), DRA006684for Japan, ERP008729 for Austria, PRJNA663646 for China (Hainan), SRP136711for Italy, PRJNA400072for IBD-UC and PRJNA400072for IBD-CD.

## ACKNOWLEDGMENTS

This research was supported by the Key Research and Development Project of Hainan Province (No. ZDYF2019150).

The study was designed by J.Z., and S.S. The experiment was performed by S.S., D.H., and C.M. Data collection was performed by S.J., and C.M. Data analysis was performed by S.S., D.H., and C.M. The manuscript was written by J.Z., and S.S. All authors read and approved the final manuscript.

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
