## [Reviewer comments · Microbiology Spectrum]

Microbiology Spectrum

Expanding the colorectal cancer biomarkers based on the human gut phageome

Siyuan Shen, Dongxue Huo, Chenchen Ma, Shuaiming Jiang, and Jiachao Zhang

Corresponding Author(s): Jiachao Zhang, Hainan University

Review Timeline:

Submission Date:	April 21, 2021
Editorial Decision:	July 25, 2021
Revision Received:	August 27, 2021
Editorial Decision:	October 12, 2021
Revision Received:	October 22, 2021
Editorial Decision:	October 25, 2021
Revision Received:	November 12, 2021
Accepted:	November 19, 2021

Editor: Zhenjiang Xu

Reviewer(s): Disclosure of reviewer identity is with reference to reviewer comments included in decision letter(s). The following individuals involved in review of your submission have agreed to reveal their identity: Sun Zheng (Reviewer #1)

Transaction Report:

DOI: <https://doi.org/10.1128/Spectrum.00090-21>

July 25, 2021

Prof. Jiachao Zhang
Hainan University
Food Science
58 renmin road
Haikou, Hainan 570228
China

Re: Spectrum00090-21 (**Expanding the colorectal cancer biomarkers based on human gut phageome**)

Dear Prof. Jiachao Zhang:

Thank you for submitting your manuscript to Microbiology Spectrum. When submitting the revised version of your paper, please provide (1) point-by-point responses to the issues raised by the reviewers as file type "Response to Reviewers," not in your cover letter, and (2) a PDF file that indicates the changes from the original submission (by highlighting or underlining the changes) as file type "Marked Up Manuscript - For Review Only". Please use this link to submit your revised manuscript - we strongly recommend that you submit your paper within the next 60 days or reach out to me. Detailed information on submitting your revised paper are below.

Link Not Available

Sincerely,

Zhenjiang Xu

Journals Department
Reviewer comments:

Reviewer #1 (Comments for the Author):

Siyuan Shen and his/her colleagues established a high-performance diagnostic model for colorectal cancer based on three training cohorts and four validation cohorts of whole metagenome sequencing data. This is an exciting work since it reveals the unique potential of the virus in the microbiome in translational medicine. Given that the newly discovered viral biomarkers for CRC can be interested in clinicians, a meta-analysis on exploring this issue should be of interest to the growing community of researchers in the field of microbiome research. I think this paper can be well improved by addressing the bellowing major concerns:

1. Compared with previous work (such as biomarkers identified in metagenome, methylation, and metabolome for CRC), the author needs to fully discuss the unique value of using phageome to discover biomarkers.
2. The Random Forest is the only method used for model construction in this paper, making the readers wonder what the performance will be if other machine learning methods are involved. Plus, the details (e.g., the importance of each biomarker and how the five biomarkers are selected) for the model construction are insufficient, especially when the whole study focuses on reporting the diagnostic model.

3. Biological interpretation of the viral biomarkers or their associations with previous studies are highly warranted to enrich the context and offer a comprehensive story to the readers.

4. The paper can be highly improved by polishing the English writing, and the authors should have this paper reviewed by someone with expertise in technical English editing before resubmission.

Minor comments for abstract and introduction:

Title: "the human gut phageome" is more in line with grammatical habits.

Line 21: "With the increasing proportion of colorectal cancer ..." should be "With the increasing prevalence of colorectal cancer ...".

Line 22: "Previous studies have highlighted the importance of bacteriophages in gastrointestinal diseases, suggesting the potential value of gut phageome in early CRC diagnostic." The logic is not clear, please revise.

Line 25: This sentence may go as "based on 317 metagenomic samples of three discovery cohorts collected from China (Hong Kong), Austria, and Japan ...".

Line 30: It should be "across different populations" without "the".

Line 32: "Meanwhile" can be replaced by "Moreover".

Line 36: "Excitingly, we observed that the five CRC-enriched phage markers also exhibited high discrimination in UC (AUC = 80.03%) and CD (AUC = 71.76%)" is confusing, the biomarker can discriminate CRC vs. other types of samples in UC and CD cohorts instead of classifying UC and CD.

Line 37: Can be easier to understand as: "The present research expands the potential of microbial biomarkers in CRC diagnosis by building a more accurate classification model based on the human gut phageome, providing a new perspective for CRC gut phagotherapy."

Line 42: "Phages are strictly host-specific, but phage biomarkers for ..." the logic here is not clear.

Line 45: Prediction model or diagnostic model? Based on the cohorts used in this paper, I did not see any possibility of building a prediction model for CRC.

Line 51-53: Duplicated with lines 36-38 in the abstract.

Line 60: The argument is too strong and doesn't fit the fact, please revise.

Line 61: Should be revised as "The individuals of bacteriophage are more than bacteria in microbiota ..."

Line 65: "Unfortunately" can be replaced with "However".

Line 69: Delete "complete".

Line 71: Please specify "them".

Reviewer #2 (Comments for the Author):

The study took the public datasets and re-analyzed for bacteriophage and found distinctive phage signatures between control and CRC fecal samples, which could also separate IBD samples with decent accuracy. The study is certainly useful and very interesting. However, I also have some major concerns:

1. the results are thin. The whole manuscript has only a single figure. There are much more analyses and content to dig. Some are listed in the below comments.

2. The description of the methods are too abridged and confusing. For example, the authors listed formula on how to compute relative abundance of gene *i* (what gene??), but how it is related to phage abundance? And there is no mention of classification methods and (hyper)parameters in Methods section? and what features and cross validation strategy was used?

3. The phage identified are very interesting. Many of them are from *F. nuleatum*, *Parvimonas micra*, and *Peptacetobacter hiranonis*. These are reported bacterial biomarkers. How good is the classification accuracy using these phages when compared to the one using these bacterial biomarkers? If the phage profile only surrogates their bacterial host profile, can the authors state the value of phage in the purpose of biomarkers?

3. what about the other phages? Little are mentioned and discussed.

4. are these phages in latent form (prophage) or lytic phage? can this be bioinformatically analyzed?

5. The CRC vs. UC/CD can be batch effects or other sequencing artifacts, because CRC and UC/CD are from different cohorts/studies. As we know, different sample processing can create big difference in data.

Staff Comments:

Preparing Revision Guidelines

For complete guidelines on revision requirements, please see the Instructions to Authors at [link to page]. **Submissions of a paper that does not conform to Microbiology Spectrum guidelines will delay acceptance of your manuscript.**

Please return the manuscript within 60 days; if you cannot complete the modification within this time period, please contact me. If you do not wish to modify the manuscript and prefer to submit it to another journal, please notify me of your decision immediately so that the manuscript may be formally withdrawn from consideration by Microbiology Spectrum.

If you would like to submit an image for consideration as the Featured Image for an issue, please contact Spectrum staff.

Comments for Manuscript number: Spectrum00090-21

Title: " Expanding the colorectal cancer biomarkers based on the human gut phageome "

Reviewer comments:

Reviewer #1 (Comments for the Author):

Siyuan Shen and his/her colleagues established a high-performance diagnostic model for colorectal cancer based on three training cohorts and four validation cohorts of whole metagenome sequencing data. This is an exciting work since it reveals the unique potential of the virus in the microbiome in translational medicine. Given that the newly discovered viral biomarkers for CRC can be interested in clinicians, a meta-analysis on exploring this issue should be of interest to the growing community of researchers in the field of microbiome research. I think this paper can be well improved by addressing the bellowing major concerns:

Response: We appreciate the reviewer's insightful comments which allowed us to improve the manuscript. Please find our point-to-point responses below.

1. Compared with previous work (such as biomarkers identified in metagenome, methylation, and metabolome for CRC), the author needs to fully discuss the unique value of using phageome to discover biomarkers.

Response: We appreciate your very insightful concern. The discovery of phage biomarkers has a unique value compared to biomarkers identified in metagenome, methylation, and metabolome for CRC. By associating the composition of intestinal phages with the occurrence of inflammatory bowel disease, Jason M. Norman and his / her colleagues found that the number and diversity of bacteria in the intestines of patients with inflammatory bowel disease were low. However, the number, abundance and diversity of some phages are increased. This indicates that this change of phage is independent of the change of bacteria [1]. At the same time, the number of bacteria in human intestine is about 10^{14} , while the number of bacteriophages is about 10^{15} - 10^{16} , which may indicate that bacteriophages have a high impact on human intestine. Compared with the biomarkers identified in the macrogenome, methylation and metabolome of CRC, the phage biomarkers we found have higher accuracy, which can reach 86.16%.

[1] Jason M. Norman, Scott A. Handley, Miles Parkes, Herbert W. Virgin. *Disease-Specific Alterations in the Enteric Virome in Inflammatory Bowel Disease*[J].

2. The Random Forest is the only method used for model construction in this paper, making the readers wonder what the performance will be if other machine learning methods are involved. Plus, the details (e.g., the importance of each biomarker and how the five biomarkers are selected) for the model construction are insufficient, especially when the whole study focuses on reporting the diagnostic model.

Response: Thanks for your query. The random forest model is the only method used to build the model in this paper. Because the random forest model has great advantages over other algorithms in many current data sets. Therefore, we choose the random forest model as the method of constructing the model in this paper. Our five CRC biomarkers were found by using both the random forest model and the Wilcoxon rank sum test. We admit that our previous description of the details of model construction and biomarker selection in materials and methods is not detailed enough, so we improve this part. The improvement contents of materials and methods are as follows:

Materials and methods

Sequence data collection

Fecal metagenomic data for CRC and control were collected for the meta-analysis. For discovery cohorts, raw SRA files and sample information were downloaded from NCBI. In the NCBI, accession of China (Hong Kong)¹ is PRJEB10878, CRC, n = 74; Control, n = 54. In the NCBI, accession of Japan² is DRA006684, CRC, n = 40; Control, n = 40. In the NCBI, accession of Austria³ is ERP008729, CRC, n = 46; Control, n = 63 (Table 1). For validation cohorts, raw SRA files and sample information were downloaded from NCBI. In the NCBI, accession of China (Hainan)⁴ is PRJNA663646, CRC, n = 8; Control, n = 12. In the NCBI, accession of Italy⁵ is SRP136711, CRC, n = 32; Control, n = 28 (Table 1). At the same time, the SRA files and sample information for the other disease validation cohorts we used were also downloaded from NCBI. Other disease cohorts included ulcerative colitis⁶ (UC, n = 76), which has been Accession PRJNA400072, and Crohn's disease⁶ (CD, n = 88), which has been Accession PRJNA400072 (Table 1).

Data quality control and phage database acquisition

Whole-genome shotgun sequencing of the samples from all cohorts was carried out on Illumina HiSeq 2000/2500 platform with similar sequencing depths. The abundances of all samples were determined by aligning the reads to the Gut Phage Database⁷ (The Gut Phage Database we used is a database of 142809 human intestinal phage genomes obtained by analyzing 28060 human

intestinal metagenomes and 2898 reference genomes of intestinal bacteria around the world. The database is linked to: http://ftp.ebi.ac.uk/pub/databases/metagenomics/genome_sets/gut_phage_database/ using Bowtie2⁸. Subsequently, for any sample N, we calculated the relative abundance as follows:

Step : Calculation of relative abundance of phages in sample N

$$a_i = \frac{b_i}{\sum_i b_i}$$

a_i : the relative abundance of phages in sample N

b_i : the number of mapped reads of phage i from sample N

Selection of bacteriophage biomarkers and application of machine learning

Five CRC bacteriophage biomarkers were identified using the random forest (RF) model⁹ and the Wilcoxon Rank-sum test. We used the random forest model to search for biomarkers from 142809 intestinal phages, and applied the R package "Ranger" (V0.12.1) to implement the random forest algorithm for each classification task. All the hyperparameters were set as default except for the number of trees set to 5000. The predictive performance of the RF model was evaluated by the cross-validation method ten-fold, and five bacteriophage biomarkers with the contribution rate >0.1% were identified. At the same time, we used the Wilcoxon Rank-sum test to search for phages with significant difference ($p < 0.0001$) between CRC patients and healthy people in three discovery cohorts, and combined analysis was performed on the differential phages found in three discovery cohorts. Five phages were found that were significantly different in all three cohorts and were enriched in the intestinal tract of CRC patients. Interestingly, the five biomarkers found by the random forest model were the same as the five biomarkers found by the Wilcoxon Rank-sum test. Therefore, we set these five phages as biomarkers, and their AUC reached 86.16.

Statistics statement

All statistical analyses were performed using R software. Vioplot was shown by the "vioplot" package. PCOA analysis was performed using the "ade4" package in R. The differential abundances of various profiles were tested with the Wilcoxon rank-sum test and were considered significantly different at $p < 0.05$. Boxplot was shown by the "ggplot2" package. Receiver operator characteristic (ROC) analysis was used to assess the performance of the microbial biomarkers using the "pROC" package in R. The Venn diagram was built using an online tool called "Omicstudio" .

1. Coker, O.O., Nakatsu, G., Dai, R.Z., Wu, W.K.K., Wong, S.H., Ng, S.C., Chan, F.K.L., Sung,

- J.J.Y., Yu, J., 2019. Enteric fungal microbiota dysbiosis and ecological alterations in colorectal cancer. *Gut* 68, 654-662.
2. Erawijantari PP, Mizutani S, Shiroma H, Shiba S, Nakajima T, Sakamoto T, Saito Y, Fukuda S, Yachida S, Yamada T, et al. Influence of gastrectomy for gastric cancer treatment on faecal microbiome and metabolome profiles. *Gut*. 2020;69:1404–1415. doi:10.1136/gutjnl-2019-319188.
 3. Feng Q, Liang S, Jia H, Stadlmayr A, Tang L, Lan Z, Zhang D, Xia H, Xu X, Jie Z, et al. Gut microbiome development along the colorectal adenoma-carcinoma sequence. *Nat Commun*. 2015;6:6528. doi:10.1038/ncomms7528.
 4. Ma, C., Chen, K., Wang, Y., Cen, C., Zhai, Q., Zhang, J., 2021. Establishing a novel colorectal cancer predictive model based on unique gut microbial single nucleotide variant markers. *Gut Microbes* 13, 1-6.
 5. Thomas AM, Manghi P, Asnicar F, Pasolli E, Armanini F, Zolfo M, Beghini F, Manara S, Karcher N, Pozzi C, et al. Metagenomic analysis of colorectal cancer datasets identifies cross-cohort microbial diagnostic signatures and a link with choline degradation. *Nat Med*. 2019;25(4):667–678. doi:10.1038/s41591-019-0405-7.
 6. Qin J, Li Y, Cai Z, Li S, Zhu J, Zhang F, Liang S, Zhang W, Guan Y, Shen D, et al. A metagenome-wide association study of gut microbiota in type 2 diabetes. *Nature*. 2012;490(7418):55–60. doi:10.1038/nature11450.
 7. Camarillo-Guerrero, L.F., Almeida, A., Rangel-Pineros, G., Finn, R.D., Lawley, T.D., 2021. Massive expansion of human gut bacteriophage diversity. *Cell* 184, 1098-1109 e1099.
 8. Langmead, B., Trapnell, C., Pop, M. & Salzberg, S. L. Ultrafast and memory-efficient alignment of short DNA sequences to the human genome. *Genome Biol* 10, R25, doi:10.1186/gb-2009-10-3-r25 (2009).
 9. Knights D, Costello EK, Knight R. Supervised classification of human microbiota. *FEMS Microbiol Rev*. 2011;35:343 – 59.

3. Biological interpretation of the viral biomarkers or their associations with previous studies are highly warranted to enrich the context and offer a comprehensive story to the readers.

Response: Thank you for your concern. Studies have shown that the number of bacteriophages is higher than that of bacteria in the intestines of patients with inflammatory bowel disease, and the changes of bacteriophages in the intestines are independent of the changes of bacteria. Besides, some studies show that the virome is a candidate for contributing to, or being a biomarker for, human inflammatory bowel disease and speculate that the enteric virome may play a role in other diseases [1]. According to the conclusions of this literature, we compared the largest phage library to find five CRC biomarkers with high accuracy.

[1] Jason M. Norman, Scott A. Handley, Miles Parkes, Herbert W. Virgin. *Disease-Specific Alterations in the Enteric Virome in Inflammatory Bowel Disease*[J]. doi:10.1016/j.cell.2015.01.002.

4. The paper can be highly improved by polishing the English writing, and the authors should have this paper reviewed by someone with expertise in technical English editing before resubmission.

Response: Thank you for your insightful comment. We've used the software Grammar to polish it. I believe the article will be improved to a certain extent.

Minor comments for abstract and introduction:

Title: "the human gut phageome" is more in line with grammatical habits.

Response: Thanks for your concern, we have modified this point per your suggestion.

Line 21: "With the increasing proportion of colorectal cancer ..." should be "With the increasing prevalence of colorectal cancer ...".

Response: Thanks for your concern, we have modified this point per your suggestion.

Line 22: "Previous studies have highlighted the importance of bacteriophages in gastrointestinal diseases, suggesting the potential value of gut phageome in early CRC diagnostic." The logic is not clear, please revise.

Response: Thanks for your concern, we have modified this point per your suggestion.

Line 25: This sentence may go as "based on 317 metagenomic samples of three discovery cohorts collected from China (Hong Kong), Austria, and Japan ...".

Response: Thanks for your concern, we have modified this point per your suggestion.

Line 30: It should be "across different populations" without "the".

Response: Thanks for your concern, we have modified this point per your suggestion.

Line 32: "Meanwhile" can be replaced by "Moreover".

Response: Thanks for your concern, we have modified this point per your suggestion.

Line 36: "Excitingly, we observed that the five CRC-enriched phage markers also exhibited high discrimination in UC (AUC = 80.03%) and CD (AUC = 71.76%)" is confusing, the biomarker can discriminate CRC vs. other types of samples in UC and CD cohorts instead of classifying UC and CD.

Response: Thanks for your concern. What we want to express is that the five phage biomarkers we found are highly accurate and specific for distinguishing between CRC patients, not only healthy people and CRC patients, but also CRC patients and IBD patients (we believe that CRC and IBD have a certain degree of similarity).

Line 37: Can be easier to understand as: "The present research expands the potential of microbial biomarkers in CRC diagnosis by building a more accurate classification model based on the human gut phageome, providing a new perspective for CRC gut phagotherapy."

Response: Thanks for your concern, we have modified this point per your suggestion.

Line 42: "Phages are strictly host-specific, but phage biomarkers for ..." the logic here is not clear.

Response: Thank you for your comment. This sentence is really not clear enough in our logic. We have changed it to: Phages are strictly host-specific, and this specificity makes them more accurate as biomarkers, but phage biomarkers for colorectal cancer have not been thoroughly explored.

Line 45: Prediction model or diagnostic model? Based on the cohorts used in this paper, I did not see any possibility of building a prediction model for CRC.

Response: Thanks for your query. The prediction model can be divided into diagnostic model, prognostic model and disease occurrence model. I think we belong to the diagnostic model of the predictive model.

Line 51-53: Duplicated with lines 36-38 in the abstract.

Response: Thanks for your concern, we have modified this point per your suggestion.

Line 60: The argument is too strong and doesn't fit the fact, please revise.

Response: Thanks for your concern, we have modified this point per your suggestion.

Line 61: Should be revised as "The individuals of bacteriophage are more than bacteria in microbiota ..."

Response: Thanks for your concern, we have modified this point per your suggestion.

Line 65: "Unfortunately" can be replaced with "However".

Response: Thanks for your concern, we have modified this point per your suggestion.

Line 69: Delete "complete".

Response: Thanks for your concern, we have modified this point per your suggestion.

Line 71: Please specify "them".

Response: Thanks for your concern, we have modified this point per your suggestion.

Reviewer #2 (Comments for the Author):

The study took the public datasets and re-analyzed for bacteriophage and found distinctive phage signatures between control and CRC fecal samples, which could also separate IBD samples with decent accuracy. The study is certainly useful and very interesting. However, I also have some major concerns:

Response: We appreciate the reviewer's insightful comments which allowed us to improve the manuscript. Please find our point-to-point responses below.

1. the results are thin. The whole manuscript has only a single figure. There are much more analyses and content to dig. Some are listed in the below comments.

Response: Thanks for your concern. I admit that our article is short and there is only one picture. This is because the submission section is Observations, which requires no more than 1200 words, no more than 2 figures/tables (our paper includes one figure and one table), and no more than 25 references. We will respond to the comments below point by point.

2. The description of the methods are too abridged and confusing. For example, the authors listed formula on how to compute relative abundance of gene i (what gene??), but how it is related to phage abundance? And there is no mention of classification methods and (hyper)parameters in Methods section? and what features and cross validation strategy was used?

Response: We appreciate your insightful comment and acute insight! We are very grateful for your question on materials and methods, so that we can better improve this part. Here are the corrections we made to the materials and methods section:

Materials and methods

Sequence data collection

Fecal metagenomic data for CRC and control were collected for the meta-analysis. For discovery cohorts, raw SRA files and sample information were downloaded from NCBI. In the NCBI, accession of China (Hong Kong)¹ is PRJEB10878, CRC, $n = 74$; Control, $n = 54$. In the NCBI, accession of Japan² is DRA006684, CRC, $n = 40$; Control, $n = 40$. In the NCBI, accession of Austria³ is ERP008729, CRC, $n = 46$; Control, $n = 63$ (Table 1). For validation cohorts, raw SRA files and sample information were downloaded from NCBI. In the NCBI, accession of China (Hainan)⁴ is PRJNA663646, CRC, $n = 8$; Control, $n = 12$. In the NCBI, accession of Italy⁵ is SRP136711, CRC, $n = 32$; Control, $n = 28$ (Table 1). At the same time, the SRA files and sample information for the other disease validation cohorts we used were also downloaded from NCBI. Other disease cohorts included ulcerative colitis⁶ (UC, $n = 76$), which has been Accession PRJNA400072, and Crohn's disease⁶ (CD, $n = 88$), which has been Accession PRJNA400072 (Table 1).

Data quality control and phage database acquisition

Whole-genome shotgun sequencing of the samples from all cohorts was carried out on Illumina HiSeq 2000/2500 platform with similar sequencing depths. The abundances of all samples were determined by aligning the reads to the Gut Phage Database⁷ (The Gut Phage Database we used is a database of 142809 human intestinal phage genomes obtained by analyzing 28060 human intestinal metagenomes and 2898 reference genomes of intestinal bacteria around the world. The database is linked to: http://ftp.ebi.ac.uk/pub/databases/metagenomics/genome_sets/gut_phage_database/) using Bowtie2⁸. Subsequently, for any sample N, we calculated the relative abundance as follows:

Step : Calculation of relative abundance of phages in sample N

$$a_i = \frac{b_i}{\sum_i b_i}$$

a_i : the relative abundance of phages in sample N

b_i : the number of mapped reads of phage i from sample N

Selection of bacteriophage biomarkers and application of machine learning

Five CRC bacteriophage biomarkers were identified using the random forest (RF) model⁹ and the Wilcoxon Rank-sum test. We used the random forest model to search for biomarkers from 142809 intestinal phages, and applied the R package "Ranger" (V0.12.1) to implement the random forest algorithm for each classification task. All the hyperparameters were set as default except for the number of trees set to 5000. The predictive performance of the RF model was evaluated by the cross-validation method ten-fold, and five bacteriophage biomarkers with the contribution rate >0.1% were identified. At the same time, we used the Wilcoxon Rank-sum test to search for phages with significant difference ($p < 0.0001$) between CRC patients and healthy people in three discovery cohorts, and combined analysis was performed on the differential phages found in three discovery cohorts. Five phages were found that were significantly different in all three cohorts and were enriched in the intestinal tract of CRC patients. Interestingly, the five biomarkers found by the random forest model were the same as the five biomarkers found by the Wilcoxon Rank-sum test. Therefore, we set these five phages as biomarkers, and their AUC reached 86.16.

Statistics statement

All statistical analyses were performed using R software. Vioplot was shown by the "vioplot" package. PCOA analysis was performed using the "ade4" package in R. The differential abundances of various profiles were tested with the Wilcoxon rank-sum test and were considered significantly different at $p < 0.05$. Boxplot was shown by the "ggplot2" package. Receiver operator characteristic (ROC) analysis was used to assess the performance of the microbial biomarkers using the "pROC" package in R. The Venn diagram was built using an online tool

called “Omicstudio” .

10. Coker, O.O., Nakatsu, G., Dai, R.Z., Wu, W.K.K., Wong, S.H., Ng, S.C., Chan, F.K.L., Sung, J.J.Y., Yu, J., 2019. Enteric fungal microbiota dysbiosis and ecological alterations in colorectal cancer. *Gut* 68, 654-662.
11. Erawijantari PP, Mizutani S, Shiroma H, Shiba S, Nakajima T, Sakamoto T, Saito Y, Fukuda S, Yachida S, Yamada T, et al. Influence of gastrectomy for gastric cancer treatment on faecal microbiome and metabolome profiles. *Gut*. 2020;69:1404–1415. doi:10.1136/gutjnl-2019-319188.
12. Feng Q, Liang S, Jia H, Stadlmayr A, Tang L, Lan Z, Zhang D, Xia H, Xu X, Jie Z, et al. Gut microbiome development along the colorectal adenoma-carcinoma sequence. *Nat Commun*. 2015;6:6528. doi:10.1038/ncomms7528.
13. Ma, C., Chen, K., Wang, Y., Cen, C., Zhai, Q., Zhang, J., 2021. Establishing a novel colorectal cancer predictive model based on unique gut microbial single nucleotide variant markers. *Gut Microbes* 13, 1-6.
14. Thomas AM, Manghi P, Asnicar F, Pasolli E, Armanini F, Zolfo M, Beghini F, Manara S, Karcher N, Pozzi C, et al. Metagenomic analysis of colorectal cancer datasets identifies cross-cohort microbial diagnostic signatures and a link with choline degradation. *Nat Med*. 2019;25(4):667–678. doi:10.1038/s41591-019-0405-7.
15. Qin J, Li Y, Cai Z, Li S, Zhu J, Zhang F, Liang S, Zhang W, Guan Y, Shen D, et al. A metagenome-wide association study of gut microbiota in type 2 diabetes. *Nature*. 2012;490(7418):55–60. doi:10.1038/nature11450.
16. Camarillo-Guerrero, L.F., Almeida, A., Rangel-Pineros, G., Finn, R.D., Lawley, T.D., 2021. Massive expansion of human gut bacteriophage diversity. *Cell* 184, 1098-1109 e1099.
17. Langmead, B., Trapnell, C., Pop, M. & Salzberg, S. L. Ultrafast and memory-efficient alignment of short DNA sequences to the human genome. *Genome Biol* 10, R25, doi:10.1186/gb-2009-10-3-r25 (2009).
18. Knights D, Costello EK, Knight R. Supervised classification of human microbiota. *FEMS Microbiol Rev*. 2011;35:343 - 59.

3. The phage identified are very interesting. Many of them are from *F. nuleatum*, *Parvimonas micra*, and *Peptacetobacter hiranonis*. These are reported bacterial biomarkers. How good is the classification accuracy using these phages when compared to the one using these bacterial biomarkers? If the phage profile only surrogates their bacterial host profile, can the authors state the value of phage in the purpose of biomarkers?

Response: Thanks for your query. The classification accuracy of the reported bacterial biomarkers can reach 80%, while the classification accuracy of the phage biomarkers we found can reach 86.16%. Bacteriophages have value as biomarkers alone. Studies have shown that the number of bacteriophages is higher than that of bacteria in the intestines of patients with inflammatory bowel disease, and the changes of bacteriophages in the intestines are independent of the changes of bacteria. Besides, some studies show that the virome is a candidate for contributing to, or being a biomarker for, human inflammatory bowel disease and speculate that the enteric virome may play a role in other diseases [1]. Therefore, we believe that phages are valuable as biomarkers.

[1] Jason M. Norman, Scott A. Handley, Miles Parkes, Herbert W. Virgin. *Disease-Specific Alterations in the Enteric Virome in Inflammatory Bowel Disease*[J]. doi:10.1016/j.cell.2015.01.002.

3. what about the other phages? Little are mentioned and discussed.

Response: Thanks for your concern. Our five CRC phage biomarkers were found simultaneously using the random forest model and the Wilcoxon rank-sum test. Among them, we used the random forest model to get 100 possible biomarkers, but we only selected 5 biomarkers with the contribution rate >0.1% as our research objects, so we rarely mentioned and discussed other possible phage biomarkers. The 100 possible biomarker contributions will be provided in supplementary table 1.

4. are these phages in latent form (prophage) or lytic phage? can this be bioinformatically analyzed?

Response: Thanks for your query. Based on the data available so far, it is not clear whether these phages are prophages or lytic phages. We compared the names and related information of the phages in the database, and the database did not specify what kind of phages these phages were. To know what these phages are, we need high-quality reads and bins of next-generation sequencing to be able to solve this problem, but our data are now obtained using next generation sequencing. So unfortunately, we don't know exactly what these phages are.

5. The CRC vs. UC/CD can be batch effects or other sequencing artifacts, because CRC and UC/CD are from different cohorts/studies. As we know, different sample processing can create big difference in data.

Response: Thanks for your concern. We acknowledge that different sample processing will indeed produce large data differences. However, the results of our meta-analysis of CRC vs. UC/CD are relatively reliable for the following four reasons: (1)The CRC sample, UC sample and CD sample data we used were all sequenced by whole genome shotgun sequencing on Illumina HiSeq 2000/2500 (Illumina, San Diego, USA) platform. (2)We carried out strict standardized quality control for all data used in meta-analysis. (3)We analyze and process CRC data, UC data and CD data in the same way. (4)We conducted cluster analysis on CRC cohort, UC cohort and CD cohort, and found that there was no significant difference among the three cohorts.

October 12, 2021

Prof. Jiachao Zhang
Hainan University
Food Science
58 renmin road
Haikou, Hainan 570228
China

Re: Spectrum00090-21R1 (**Expanding the colorectal cancer biomarkers based on the human gutphageome**)

Dear Prof. Jiachao Zhang:

We carefully reviewed your responses to the comments and think a few of them are not sufficiently satisfactory. Particularly, 1) The comparison of disease classification using phage vs. bacteria. You cited the accuracies from literature, which is not a fair comparison. For completeness of the study, we encourage you to add a figure to Figure 1 to show the prediction of bacteria profile on these exact same data sets. 2) The batch effect still exists even with same sequencing platform and bioinformatic pipeline. Your PCoA plot actually shows the 3 datasets were grouped. You could try to remove the batch effects or avoid the claims that require combining all the datasets together. With these two points further addressed, your manuscript is readily accepted for MS.

Thank you for submitting your manuscript to Microbiology Spectrum. When submitting the revised version of your paper, please provide (1) point-by-point responses to the issues raised by the reviewers as file type "Response to Reviewers," not in your cover letter, and (2) a PDF file that indicates the changes from the original submission (by highlighting or underlining the changes) as file type "Marked Up Manuscript - For Review Only". Please use this link to submit your revised manuscript - we strongly recommend that you submit your paper within the next 60 days or reach out to me. Detailed information on submitting your revised paper are below.

Link Not Available

Sincerely,

Zhenjiang Xu

Journals Department
Reviewer comments:

Staff Comments:

Preparing Revision Guidelines

To submit your modified manuscript, log onto the eJP submission site at <https://spectrum.msubmit.net/cgi-bin/main.plex>. Go to Author Tasks and click the appropriate manuscript title to begin the revision process. The information that you entered when you first submitted the paper will be displayed. Please update the information as necessary. Here are a few examples of required

updates that authors must address:

Please return the manuscript within 60 days; if you cannot complete the modification within this time period, please contact me. If you do not wish to modify the manuscript and prefer to submit it to another journal, please notify me of your decision immediately so that the manuscript may be formally withdrawn from consideration by Microbiology Spectrum.

Comments for Manuscript number: Spectrum00090-21R1

Title: " Expanding the colorectal cancer biomarkers based on the human gut phageome "

Reviewer comments:

We carefully reviewed your responses to the comments and think a few of them are not sufficiently satisfactory. Particularly, 1) The comparison of disease classification using phage vs. bacteria. You cited the accuracies from literature, which is not a fair comparison. For completeness of the study, we encourage you to add a figure to Figure 1 to show the prediction of bacteria profile on these exact same data sets. 2) The batch effect still exists even with same sequencing platform and bioinformatic pipeline. Your PCoA plot actually show the 3 datasets were grouped. You could try to remove the batch effects or avoid the claims that require combining all the datasets together. With these two points further addressed, your manuscript is readily accepted for MS.

Response: We appreciate the reviewer's insightful comments which allowed us to improve the manuscript. Please find our point-to-point responses below.

1. The comparison of disease classification using phage vs. bacteria. You cited the accuracies from literature, which is not a fair comparison. For completeness of the study, we encourage you to add a figure to Figure 1 to show the prediction of bacteria profile on these exact same data sets.

Response: We appreciate your very insightful concern. We decided to adopt your suggestion and add the prediction of bacterial distribution on the same data set to the article and Figure 1. We believe that your suggestion not only made our article more complete, but also made our article more profound.

2. The batch effect still exists even with same sequencing platform and bioinformatic pipeline. Your PCoA plot actually show the 3 datasets were grouped. You could try to remove the batch effects or avoid the claims that require combining all the datasets together.

Response: Thanks for your concern. We agree to your suggestion. To eliminate batch effect completely, these samples should be resequenced on the same platform and in the same batch, but this was clearly not possible in this study. Therefore, we are more willing to adopt your second suggestion "avoid the claims that require combining all the datasets together". Therefore, we give up cluster analysis by combining three datasets together. When analyzing the specificity of phage biomarkers, we carried out strict quality control on all data, annotated the biomarkers with random forest method, and then calculated the specificity of these biomarkers with the application of an area under the receiver-operating characteristic curve.

October 25, 2021

Prof. Jiachao Zhang
Hainan University
Food Science
58 renmin road
Haikou, Hainan 570228
China

Re: Spectrum00090-21R2 (**Expanding the colorectal cancer biomarkers based on the human gutphageome**)

Dear Prof. Jiachao Zhang:

Thank you for submitting your manuscript to Microbiology Spectrum. I believe the previous two issues are not satisfactorily addressed. Please describe how the bacterial profile were computed and correct batch effects for Fig 1G/H/I. Please use this link to submit your revised manuscript - we strongly recommend that you submit your paper within the next 60 days or reach out to me. Detailed information on submitting your revised paper are below.

Link Not Available

Sincerely,

Zhenjiang Xu

Journals Department
Reviewer comments:

Staff Comments:

Preparing Revision Guidelines

Please return the manuscript within 60 days; if you cannot complete the modification within this time period, please contact me. If you do not wish to modify the manuscript and prefer to submit it to another journal, please notify me of your decision immediately so that the manuscript may be formally withdrawn from consideration by Microbiology Spectrum.

Comments for Manuscript number: Spectrum00090-21R2

Title: " Expanding the colorectal cancer biomarkers based on the human gut phageome "

Reviewer comments:

Thank you for submitting your manuscript to Microbiology Spectrum. I believe the previous two issues are not satisfactorily addressed. Please describe how the bacterial profile were computed and correct batch effects for Fig 1G/H/I.

Response: We appreciate the reviewer's insightful comments which allowed us to improve the manuscript. Please find our point-to-point responses below.

1. The comparison of disease classification using phage vs. bacteria. You cited the accuracies from literature, which is not a fair comparison. For completeness of the study, we encourage you to add a figure to Figure 1 to show the prediction of bacteria profile on these exact same data sets. (Please describe how the bacterial profile were computed.)

Response: We appreciate your very insightful concern. We decided to adopt your suggestion and add the prediction of bacterial distribution on the same data set to the article and Figure 1. The calculation method of bacterial abundance and screening method of bacterial biomarkers have been described in materials and methods. We believe that your suggestion not only made our article more complete, but also made our article more profound.

2. The batch effect still exists even with same sequencing platform and bioinformatic pipeline. Your PCoA plot actually show the 3 datasets were grouped. You could try to remove the batch effects or avoid the claims that require combining all the datasets together.(Correct batch effects for Fig 1G/H/I.)

Response: We are very grateful to you for raising the issue of batch effect, which is very helpful to us. Batch effects are unwanted data variations that may obscure biological signals, leading to bias or errors in subsequent data analyses. Effective evaluation and elimination of batch effects are necessary for omics data analysis. Therefore, PVCA method in the online tool "BatchServer" was used to evaluate the batch effect of sample data¹, and we found that the data did have a high batch effect, as shown in the figure below. So we used ComBat in the SVA package to remove batch effects from the sample data. ComBat is a widely used method of eliminating batch effects. The results did differ slightly after batch effect was removed from those without it. Methods and results of removing batch effect are presented in this paper. After removing the batch effect, the validation cohort also had high accuracy, with an AUC of 81.97%. However, for the specificity of five CRC phage biomarkers, the results changed significantly after the batch effect was removed. The specificity of 5 phage biomarkers against CRC and UC was high, and the AUC reached 78.02%. However, the 5 phage biomarkers had no specificity for CRC and CD, and the AUC was only 48.00%. Although the 5 CRC phage biomarkers are not specific for CD, this is also interesting to some extent and we decided to retain this conclusion. Finally, thank you again for your suggestions on batch effect, which sublimated our article and made the results more accurate.

1. Zhu T, Sun R, Zhang F, Chen GB, Yi X, Ruan G, et al. BatchServer: A Web Server for Batch Effect Evaluation, Visualization, and Correction. J Proteome Res 2021; 20:1079-86.

November 19, 2021

Prof. Jiachao Zhang
Hainan University
Food Science
58 renmin road
Haikou, Hainan 570228
China

Re: Spectrum00090-21R3 (**Expanding the colorectal cancer biomarkers based on the human gutphageome**)

Dear Prof. Jiachao Zhang:

Your manuscript has been accepted, and I am forwarding it to the ASM Journals Department for publication. You will be notified when your proofs are ready to be viewed.

Sincerely,

Zhenjiang Xu
Editor, Microbiology Spectrum
